# Different Parts of a *Dendrocalamus brandisii* (Munro) Kurz Shoot, Rather Than the Shoot's Height, Are More Indicative of Its Nutrient Properties

**Luxi Li, Yuzhuo Wen, Jingyun Xu, Tiandao Bai, Mei Yang and Weixin Jiang \***

Key Laboratory of National Forestry and Grassland Administration on Cultivation of Fast-Growing Timber in Central South China, Guangxi Key Laboratory of Forest Ecology and Conservation, College of Forestry, Guangxi University, Nanning 530004, China; liluxi020816@163.com (L.L.); wenyuzhuo0130@163.com (Y.W.); 15296103478@163.com (J.X.); btd@gxu.edu.cn (T.B.); fjyangmei@126.com (M.Y.)
\* Correspondence: jwx_1985@163.com; Tel.: +86-0771-3271428

**Abstract:** Bamboo shoots are considered as a healthy food and are popular in Asian cooking. The distribution of nutrients and their dynamics during the growth of bamboo shoots play a crucial role in guiding the harvesting and processing of bamboo shoots. In this study, *Dendrocalamus brandisii* (Munro) Kurz, an important bamboo species for harvesting fine edible shoots in southern China and Southeast Asia, was used to measure several indicators related to the edibility and nutritional value of fresh shoots across five height grades (H1: 20~30 cm; H2: 31~40 cm; H3: 41~50 cm; H4: 51~60 cm; and H5: 61~70 cm). The results indicated that, although the nutrient and mineral elements showed an increasing (crude fiber content, CFC), decreasing (total soluble sugars content, TSSC; ash content, AC; Fe; Zn), or fluctuating (soluble protein content, SPC; P; K; Ca) pattern with the growth of bamboo shoots, both the CFC and TSSC showed the highest values in the lower part at five growth heights, indicating that carbohydrates were mainly enriched in the bases of bamboo shoots. The SPC, AC, and other mineral elements were higher in the upper part, especially in H1–H3. Combined with the relatively high activity of metabolic enzymes (sucrose-phosphate synthase, SPS; neutral invertase, NI) in the upper part of bamboo shoots (although not statistically significant), it can be inferred that the shoot tip may be the main tissue for early nutrient synthesis and metabolism. Compared with the shoot height, different parts of a *D. brandisii* bamboo shoot are more indicative of its nutrient properties. Although all heights of bamboo shoots showed an abundance of nutrients and mineral elements, bamboo shoots with a height of less than 40 cm had a higher TSSC, AC, Fe, and Zn, and a lower CFC, thus having a better balance between nutrients and edibility.

**Keywords:** bamboo shoots; mineral elements; nutrients; enzymatic activity

## 1. Introduction

Bamboo shoots, with their rich content of nutrients, vitamins, minerals, and dietary fibers, and low content of fat and calories, are considered as a health food. They are not only used in a variety of traditional delicacies in many Asian countries, including China, India, Japan, Korea, Thailand, and the Philippines, but have also been introduced and consumed in many countries in Africa, North America, and Europe [1,2]. Shoots of many bamboo species are generally harvested after they attain certain heights aboveground [3]. However, the edibility and nutrients in the shoots of some bamboo species showed a depletion with the shoot's growth [4,5]. It is essential to reveal the rules of nutrient accumulation and distribution in bamboo shoots, to determine a suitable harvest time or optimal shoot height for a balance between nutrients and taste.

Bamboo shoots with a high content of dietary fiber have beneficial effects on lipid profiles and bowel function, and protect against many chronic diseases. Especially, the

insoluble dietary fiber (IDF) from bamboo shoots effectively suppressed the development of obesity, and modulated the diversity of the intestinal microbial population, in mice fed with a high-fat diet [6]. However, the crude fiber will also lead to an increase in intestinal burden [7]. A daily intake of 30~45 g is the most suitable for the healthy development of the human body, and the crude fiber content in bamboo shoots is just right for this need [8]. The amino acid content of bamboo shoots is much higher than that of other vegetables and the total soluble sugar content ensures that it is a healthy green food with low sugar [2]. Bamboo shoots contain a large number of minerals, which are essential for the activity of enzymes and the structure of tissues in the body [2,9]. Zn and Cu are necessary for the activity of the enzyme superoxide dismutase (SOD), and iron is also a cofactor, in that it is the most abundant trace element in the body and most of the iron is present as iron bound to proteins [10]. Although many studies reported the constituents of bamboo shoots, including proteins, starch, carbohydrates, fat, dietary fiber, vitamins, and minerals, among various bamboo species [4,11–14], the changes in nutrient components with the shoot growth of many bamboo species are still not clearly described. Little is known about the distribution of nutrient elements within bamboo shoots during their development and aging, which would probably be useful for understanding the mechanism of nutrient metabolism and transmission in bamboo shoots.

*Dendrocalamus brandisii* (Munro) Kurz, also known as sweet bamboo, is a large tropical and subtropical evergreen sympodial bamboo species in the subfamily Bambusoideae (Gramineae). It originates from southern and northeastern India and Myanmar, and it is mainly distributed in southern China (specifically, the Yunnan province), as well as in South and Southeast Asian countries, and Ethiopia, Bosnia, Herzegovina, and Brazil. It can be cultivated to harvest edible bamboo shoots and bamboo culms for use as a building material (http://www.iplant.cn/info/ Dendrocalamus%20brandisii?t=z) (accessed on 20 January 2024). In particular, the shoot of *D. brandisii* remains crisp and tender after emerging from the ground up to 50 cm and beyond. Given its rapid growth and high economic benefits, this species of bamboo has become a key focus of development in Yunnan province and has also been introduced to other provinces in southern China, such as Guangxi, Guangdong, Fujian, and Zhejiang [15,16]. Early studies on the nutritional composition of *D. brandisii* revealed a low crude fat content (0.281%) and a complete set of amino acids, totaling 17 and amounting to 227.64 mg·g$^{-1}$. These amino acids could supplement the essential ones that cannot be synthesized by the human body [17,18]. Compared with other *Dendrocalamus* species, *D. brandisii* exhibited excellent nutritional quality, characterized by the highest ash and total amino acid content, but the lowest crude fiber and fat content [17]. Notably, those results mainly derived from one developmental point or stage; no study has focused on the distribution of nutrient components along the bamboo shoots and how this distribution changes with shoot height growth. During the growth of bamboo shoots, do the nutrient components keep stable or show different trends of variation in different parts of the bamboo shoots? In this study, *D. brandisii* shoots with five grades of growth heights, and three parts of the bamboo shoots, were studied to (1) explore the quantities and distribution of nutrient components during the growth of bamboo shoots within the edible range, and (2) identify the optimal height for harvesting bamboo shoots to obtain high-quality produce.

## 2. Materials and Methods

### 2.1. Sample Collection

*Dendrocalamus brandisii* (Munro) Kurz shoots were harvested from an experimental understory bamboo cultivation site at the Guangxi State-Owned Qipo Forest Farm in Guangxi, China (108.21° E, 22.68° N, Altitude 200~300 m) in 30 July 2022. The experimental forest was planted in March 2018 and flourished naturally with minimal human intervention. The soil at the experimental site consists of lateritic red earth, and the physical and chemical properties of the soil before bamboo planting have been previously reported

[19]. The mean annual temperature is 22.0 °C, with an average maximum of 26 °C and a minimum of 18 °C. The coldest month is January, and the hottest months are July and August. The annual average solar radiation is 4611.3~4670.9 MJ·m⁻², the accumulated temperature of ≥10 °C is 5000~8300 °C, and the sunshine hours are more than 1800 hours throughout the year. The mean annual precipitation is 1200~1300 mm, with a relative humidity of 70~80%, which is suitable for the growth of clumping bamboo species.

The shoots of *D. brandisii* were harvested on 30 July 2022, which was at the flourishing stage of the bamboo shooting. Fresh and healthy juvenile *D. brandisii* shoots emerging 20~70 cm above the ground were harvested in the morning hours, because the transpiration is lower. The shoots were measured and divided into five groups based on their height above the ground: H1 (20~30 cm), H2 (31~40 cm), H3 (41~50 cm), H4 (51~60 cm), and H5 (61~70 cm) (Figure 1). Three shoots were harvested in each group, and a total of 15 healthy shoots (with a ground diameter of about 12~14 cm) were harvested for further measurement and analysis. The edible portions of the bamboo shoots that were less than 20 cm in height above the soil were small and short, and were not considered due to their low yield and effectiveness in shoot production. The length, diameter at the base of the shoot, and weight of the 15 fresh shoots were recorded before and after removing the sheaths. After peeling off the sheaths, each bamboo shoot was divided into three parts: top, middle, and base (Figure 1). Each part was cut into many similar cubes, each measuring 1.5 cm in length, 1.5 cm in width, and 1 cm in thickness. A total of 45 samples were collected, and 5 g of each sample was stored in vacuum-sealed plastic bags at −80 °C for an analysis of nutrient-related enzyme activity. The remaining portion of each sample was exposed to 130 °C for 30 min, then baked in an oven at 70 °C until a constant weight was achieved. The samples were then powdered using tissue grinders (Tissuelyer-192, Shanghai Jingxin Industrial Development Co., Ltd., Shanghai, China) for subsequent nutritional composition analysis.

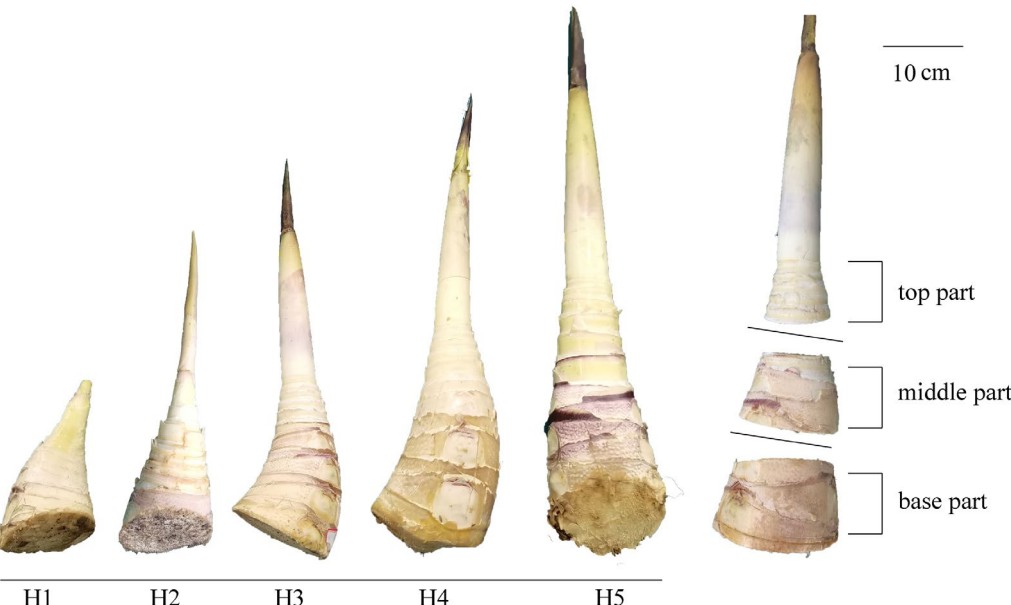

**Figure 1.** Peeled fresh bamboo shoots of *D. brandisii* at five height grades and three parts within a shoot.

To explore the status of, and changes in the mineral components in, the soil of the experiment stand before and after bamboo shoots emerging, about 1.5~2.0 kg of soil in the root zone of the bamboo was collected on 10 March and 30 July 2022, in the early stage and flourishing stage of the bamboo shoots emerging, respectively. Six sites (replications)

in the bamboo stand were randomly selected and each soil sample mixed from the root zones of five adjacent bamboo bushes.

### 2.2. Nutrient and Mineral Indicators Measurement

The moisture content (MC) was determined by calculating the percentage weight loss during drying, following the Chinese standard GB 5009.3-2003) [20]. Crude fiber content (CFC) was determined using the acid-base washing method and expressed as g per 100 g dry weight (DW) according to the PRC [21]. The total soluble sugars content (TSSC) was determined using traditional anthrone colorimetry [22], and the soluble protein content (SPC) was measured using a Coomassie Brilliant Blue G-250-based assay [22]. The ash content (AC) was determined through muffle furnace combustion at 550 °C for 4 h, following the Chinese standard GB/T 5009.4-2003 (GB/T, National Standards of the People's Republic of China) [23]. During the measurement process, each sample was technically repeated three times to calculate a mean value.

For the estimation of essential mineral elements such as calcium (Ca), iron (Fe), potassium (K), and zinc (Zn) [24], the shoot samples underwent wet digestion using a mixture of sulfuric acid, perchloric acid, and nitric acid in a ratio of 1:4:10. The mixture was heated until the digestion material became clear. A total of 5 mL hydrochloric acid (HCl) was added, and the volume was adjusted to 100 mL with distilled water. The mineral elements in the soil sample were determined according to the standard protocol of the State Forestry Administration [25], where the Zn and Fe were extracted by 0.1 M HCl. Ca was extracted 2–3 times by 1 M $CH_3COONH_4$ (pH7.0), until there was no calcium ion reaction in the leaching solution; then, 5ml $SrC_6H_2O$ (30 g·L$^{-1}$) solution was added to eliminate the interference of other ions. The available K and total K were determined by ammonium acetate extraction and wet digestion, using a hydrofluoric acid and perchloric acid solution, respectively. All the mineral elements were determined using a flame atomic absorption spectrophotometer (FAAS, Analytikjena novAA350, Jena, Germany), at wavelengths of 213.8 nm (Zn), 248.3 nm (Fe), 776.5 nm (K), and 442.7 nm (Ca), respectively. The total phosphorus (P) was estimated using the sulfuric acid–hydrogen peroxide digestion–Mo-Sb colorimetric method, and the available P in the soil was determined by the sodium bicarbonate extraction Mo-Sb colorimetric method [25]. A quantitative determination was conducted using a spectrophotometer (Tecan, Infinite™ M200 PRO, Männedorf, Switzerland) to measure the absorbance at 700 nm.

To determine the activity of the main sugar metabolism enzymes, we used the method of Lowell et al. [26] to measure neutral invertase (NI) activity. NI activity was measured in a mixture containing 0.2 M potassium phosphate buffer, 5 mM $MgCl_2$, 0.1% β-mercaptoethanol (pH 7.5), and crude extract. The reaction was stopped at 30 min by adding 600 μL of 1% (*w/v*) 3,5-dinitrosalicylic acid (DNS) and heating it at 100 °C for 5 min, followed by measuring the absorbance at 540 nm. The activity of the invertase was defined as the amount of glucose produced per hour per kilogram of fresh weight of shoot tissue under the experimental conditions. The extraction and assay of sucrose-phosphate synthase (SPS) followed the method described by Komatsu et al. [27]. Color development was measured at 620 nm, and the SPS activities were expressed as mg Suc·h$^{-1}$·g$^{-1}$ FW. ATP synthetase (ATP) activity was measured following the method described by Xu [28]. The inorganic phosphorus content produced per milligram of chloroplast per hour was used as an activity unit of ATP (μmol (P)·mg (chl)$^{-1}$·h$^{-1}$) to indicate the magnitude of ATPase activity. All the indicators measured in this study are listed in Table 1.

**Table 1.** Indicators and their abbreviations.

| Category | Indicator (Unit) | Abbreviation |
|---|---|---|
| Nutrient and mineral element | Moisture content | MC |
| | Crude fiber content (%) | CFC |
| | Total soluble sugar content (mg·g$^{-1}$) | TSSC |

| | Soluble protein content (mg·g⁻¹) | SPC |
| | Ash content (%) | AC |
| | Total phosphorus content (mg·g⁻¹) | P |
| | Available phosphorus content (ug·g⁻¹) (soil) | AP |
| | Total potassium content (mg·g⁻¹) | K |
| | Available potassium content (ug·g⁻¹) (soil) | AK |
| | Calcium (mg·g⁻¹) | Ca |
| | Iron (mg·100 g⁻¹) | Fe |
| | Zinc (mg·100 g⁻¹) | Zn |
| Enzyme | Neutral invertase (U·g⁻¹·h⁻¹ FW) | NI |
| | Sucrose-phosphate synthase (U·g⁻¹·h⁻¹ FW) | SPS |
| | ATP synthetase (μmol (P)·mg (chl)⁻¹·h⁻¹) | ATP |

*2.3. Data Analysis*

A nested analysis of variance (ANOVA) was conducted on all the indicators among the five groups of shoots with varying heights. The three parts—namely, the upper, middle, and base parts of the shoot—within each height group were considered as a nested factor. The p-values for the ANOVA model were obtained using permutation tests instead of the normal theory, because some indicators (K, SPS, and ATP) presented non-normal distributions [29]. Spearman's correlation analysis was conducted to examine the correlation between any pair of indicators. Principal component analysis (PCA) was performed on all indicators that showed significant differences between shoot heights or shoot parts. Hierarchical clustering was conducted to reveal the differences between various sections and height grades. All statistical analyses and plot visualizations were conducted using R software version 4.2.1 [30] and its extensional packages [31,32].

**3. Results**

*3.1. Nutrients and Mineral Contents*

Significant differences ($p < 0.05$) were observed in the CFC and TSSC among various height grades and parts of the *D. brandisii* shoots (Figure 2). The CFC increased with the height grade of the bamboo shoots. The TSSC remained relatively stable from H1 to H2, and decreased significantly after H2 and then maintained a stable level at H4 and H5. The content of CFC and TSSC in the base part was higher than that in the middle and top parts. The SPC also varied significantly across height grades, with the lowest level of SPC observed at H2 in all parts. There were no significant differences in the MC and AC among the various heights and parts. However, it should be noted that the AC in the top part was slightly higher than that in other parts from H1 to H3, then decreased after H3, and then remained at the same level as that in other parts.

Three out of five mineral elements (P, K, and Fe) showed significant differences, while Zn showed marginal differences between height grades (Figure 2). All five mineral elements in the top part of the bamboo shoots showed relatively higher levels during the first three height grades (from H1 to H3), although only three elements (P, Ca, and Zn) showed statistically significant differences among parts of the shoot. It is worth noting that the elements Ca, Fe, and Zn exhibited a similar trend across the height grades, and the levels of the three elements in the top part showed a notable decline after the height grade H3.

The content of mineral elements in the root soil (Table S1) demonstrated that the Zn and available K content in the soil at the flourishing stage of the bamboo shoots were significantly lower than those at the early stage ($p < 0.05$), while the Ca and total K were opposite. The growth of the bamboo shoots had no significant effect on the P content in soil.

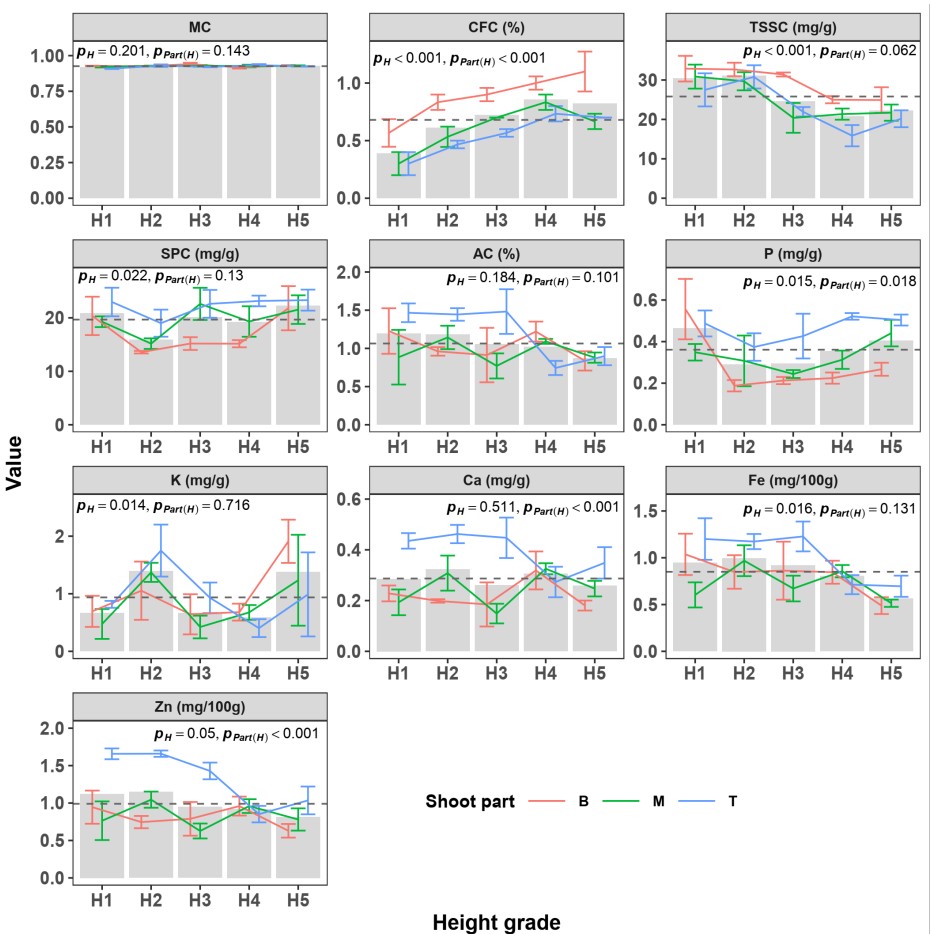

**Figure 2.** Nutrient and mineral element content in different shoot parts of *D. brandisii* at five height grades. $p_H$ is the *p*-value between different height grades, and $p_{Part(H)}$ is the *p*-value between different parts within each height grade. The horizontal dashed line is the overall mean of corresponding indicators. MC, moisture content; CFC, crude fiber content; TSSC, total soluble sugar content; SPC, soluble protein content; AC, ash content; B, base part; M, middle part; T, top part.

### 3.2. Sugar Metabolism Enzymes

No significant difference was observed in the activity of the three enzymes among the various parts and height grades of *D. brandisii* shoots. Nonetheless, the activity of the three enzymes in the basal or middle parts appears to follow a similar pattern across the five height categories. In the top part, however, the three enzymes responded differently to the increase in shoot height (Figure 3).

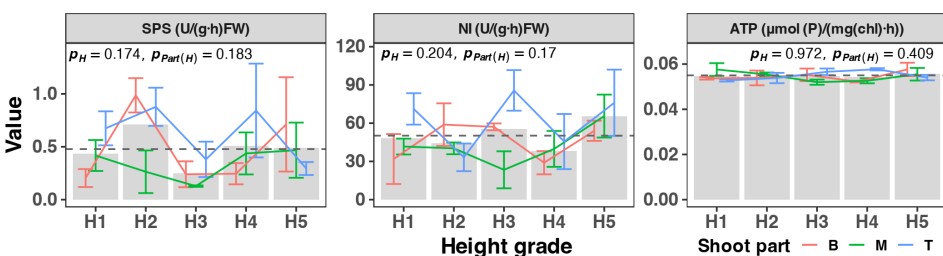

**Figure 3.** The activity of three enzymes in different shoot parts of *D. brandisii* at five height grades. pH is the *p*-value between height grades, and $p_{Part(H)}$ is the *p*-value between different parts within each height grade. The horizontal dashed line is the overall mean of corresponding indicators. SPS, sucrose-phosphate synthase; NI, neutral invertase; ATP, ATP synthetase; B, base part; M, middle part; T, top part.

*3.3. Correlation between Indicators*

A correlation analysis of the indicators in the top part (Figure 4A) revealed that the CFC was negatively correlated with most of the indicators, while the TSSC showed a positive correlation with the AC and most mineral elements (Ca, Fe, and Zn). The AC and the tested mineral elements (except P) were positively correlated, although some of the coefficients were not statistically significant. In the middle shoots, the AC was positively correlated with K, Ca, Zn and Fe, while the SPC exhibited a negative correlation with the other elements (Figure 4B). The basal shoots showed a similar correlation between AC, Ca, Fe, and Zn (Figure 4C). But notably, the SPC was significantly positive correlated with P. Overall, the CFC was negatively correlated with the SPC, P, Ca, and Zn ($p < 0.05$). The SPC was negatively correlated with the TSSC ($p < 0.001$), but positively correlated with P ($p < 0.001$). The AC exhibited a positive correlation with K, Ca, Fe, and Zn ($p < 0.01$). There were positive correlations between K, Ca, Fe, and Zn, except between K and Fe ($p < 0.05$). No significant correlation was observed between the three enzymes and all nutrient and mineral indicators, except for NI and Zn (Figure 4D).

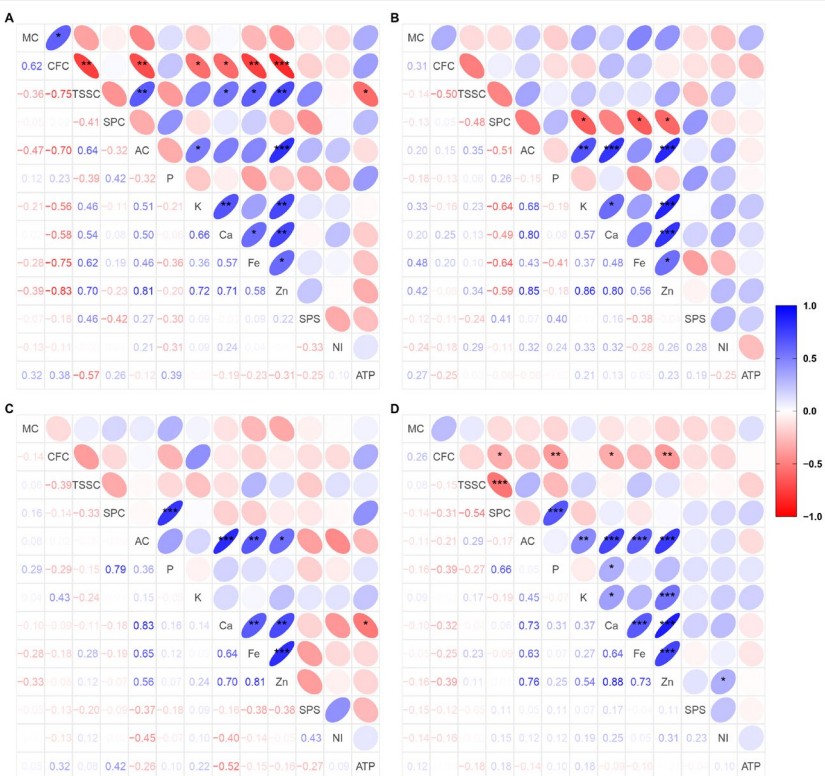

**Figure 4.** Spearman's correlation coefficients (lower triangle) and significance (upper triangle) between indicators of *D. brandisii* shoot. Plots (**A–C**) were estimated on the corresponding sub-data of the top, middle, and base parts, and plot (**D**) was estimated on the overall data. Asterisks indicate the level of significance (* $p < 0.05$, ** $p < 0.01$, and *** $p < 0.001$).

*3.4. Principal Components Analysis and Hierarchical Clustering*

The principal component analysis (PCA) revealed that the eigenvalues of the first three principal components (PCs) all exceeded one, effectively capturing a significant portion (74.61%) of the nutritional information related to the various parts and height grades of *D. brandisii* shoots (Table 2). Among these elements, Zn, Ca, Fe, and AC contributed more to PC1, while the SPC, P, and TSSC contributed more to PC2. K and CFC made a greater contribution to PC3. The PC1 in the top part of the shoots was significantly higher than that in the base and middle parts from H1 to H3, and it declined sharply from H3 to H4 (Figure 5A). PC2 exhibited a similar pattern in three parts across the different height

grades, decreasing from H1 to H2 and then gradually increasing. PC3 increased from H1 to H2 in all parts and then remained relatively stable in the upper and middle parts from H2 to H5, but again, increased in the base part after H3.

The two-dimensional plot of PC1 versus PC2 and PC3 showed that, although there was no clear distinction separation among the five height grades, a significant number of points from H4 and H5 were situated in the top-left area, whereas more points from H1 and H2 were found in the bottom-right area (Figure 5B,C). Among all height categories, a large number of points in the top part were situated in the top-right area, while the points in the middle and base parts largely overlap in the left and bottom areas (Figure 5D,E).

**Table 2.** Principal component loading for nutrients and mineral indicators of *D. brandisii* shoots at varying heights aboveground.

| Indicator | PC1 | PC2 | PC3 | PC4 | PC5 |
|---|---|---|---|---|---|
| TSSC | 0.223 | −0.696 | −0.460 | 0.328 | 0.305 |
| SPC | 0.056 | 0.894 | 0.097 | 0.106 | 0.009 |
| CFC | −0.483 | −0.256 | 0.637 | −0.328 | 0.419 |
| AC | 0.810 | −0.249 | 0.067 | −0.158 | 0.035 |
| P | 0.353 | 0.731 | −0.138 | 0.179 | 0.462 |
| K | 0.346 | −0.216 | 0.688 | 0.591 | −0.054 |
| Ca | 0.884 | 0.091 | 0.152 | −0.203 | −0.096 |
| Fe | 0.808 | −0.125 | −0.058 | −0.205 | 0.150 |
| Zn | 0.959 | 0.021 | 0.077 | −0.033 | −0.078 |
| Eigenvalue | 3.542 | 2.016 | 1.156 | 0.717 | 0.524 |
| variance percent (%) | 39.356 | 22.404 | 12.843 | 7.969 | 5.826 |

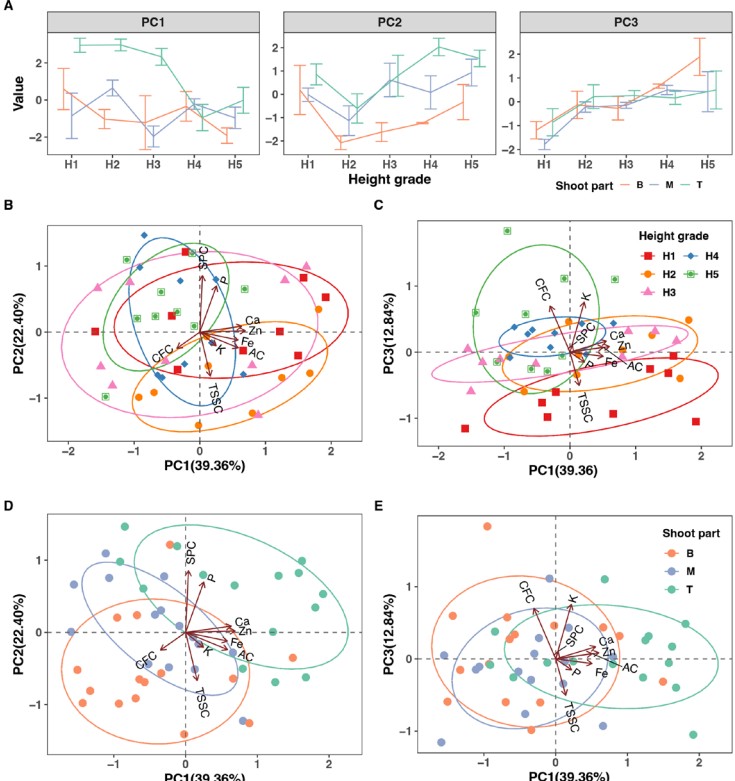

**Figure 5.** The values of the first three principal components (PCs) on three shoots part of *D. brandisii* at five height grades. The first three principal components (PC1, PC2, PC3) of the three parts in different height grades (**A**). The two-dimensional plot of PC1 versus PC2 (**B**) and PC3 (**C**) among the five height grades. The two-dimensional plot of PC1 versus PC2 (**D**) and PC3 (**E**) among the three

parts across all height categories. The shoot part B, M and T in the subfigure E and A denote the base, middle, and top part of the shoot.

All parts from the five height grades were clustered into two groups based on Euclidean distances (Figure 6). The top parts of H1, H2, and H3 were clustered into one group, while the remaining parts from the other height grades were clustered together. It should be noted that all parts in H5 were clustered into a subgroup, indicating similar nutrient properties among different parts. All indicators were clustered into six groups. Notably, the concentrations of the AC, Ca, Fe, and Zn were relatively high in the top parts of H1, H2, and H3, while the MC and CFC were more abundant in the base and middle parts of H4 and H5.

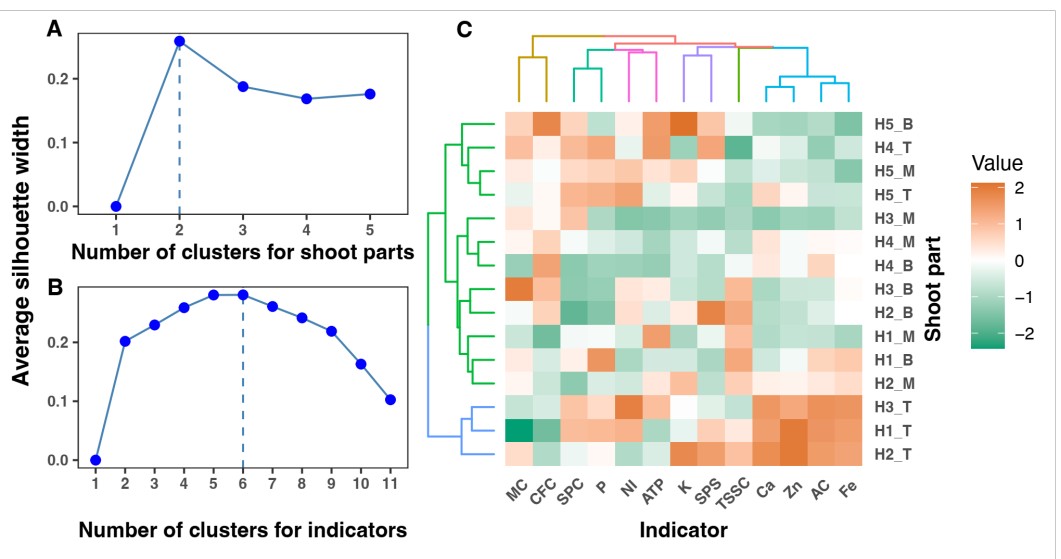

**Figure 6.** Hierarchical clustering between shoot parts. Optimal number of clusters of shoot parts of five height grades (**A**) and indicators (**B**) depends on average silhouette width. The standardized indicator values of the shoot parts in different height grades were displayed as a heat plot (**C**).

## 4. Discussion

### 4.1. The Levels of Nutrients and Mineral Components in Shoots of D. brandisii and Its Safety Considerations as Food

This study demonstrated similar values for the MC (92.6%), CFC (0.68%), and AC (1.06%) compared to previous reports on the shoots of *D. brandisii* (MC, 92.57~92.85%; CFC, 0.63~0.66%; AC, 0.76~0.89%) [17,18]. The average of the TSSC and SPC in *D. brandisii* shoots were 25.8 mg·g⁻¹ and 19.7 mg·g⁻¹, which were lower than those reported by Chongtham et al. [4] (49 mg·g⁻¹ and 23.1 mg·g⁻¹, correspondingly). The average P content (0.36 mg·g⁻¹) and K content (0.94 mg·g⁻¹) were comparable to the results reported by Hui et al. [33], but the average Ca content (0.29 mg·g⁻¹) and Zn content (1.0 mg per 100g) are much higher than those reported by the latter. We inferred that these inconsistencies may be related to the varying understory soil environments in the growing areas. Previous studies have shown that soil, altitude, and other environmental factors could affect the nutritional content of bamboo shoots [34–37], even though exhibiting no taste differences [38].

Although this study indicated that the shoot of *D. brandisii* contains high levels of Ca, Zn, and Fe, a person over 9 years old need to consume about 800 g or more bamboo shoots per day according to the World Health Organization's (WHO) recommended daily intake of calcium, iron, and zinc [39], which is almost impossible in most cases, so there is no need to worry about excessive intake. Given the trends of all nutritional indicators, the

shoots of *D. brandisii*, which can grow up to 40 cm in height, still exhibit a good nutritional quality and palatability.

### 4.2. Distribution and Dynamics of Nutrient Components during Shoot Growth of D. brandisii

Although many studies have reported changes in nutrient components as bamboo shoots age [4,5,40,41], few studies have focused on the dynamics of nutrient content in different parts of shoots during growth. We found that the nutrient response varied across different parts and did not correspond with the height growth of the bamboo shoots. The genomic analysis for the *D. brandisii* shoots verified that the upper part of the shoot appears to be more involved in cell division, while the base and middle portion of the shoot tend to be associated with cell wall biogenesis [38]. This can explain why the CFC and TSSC were most responsive at the base part of the bamboo shoots. Generally, the higher the TSSC and the lower the CFC, the better the taste of the bamboo shoots, although the CFC is always recognized as beneficial for human health [42,43]. The CFC stably increased with the height grades, aligning with findings for other bamboo shoots [17]. Conversely, the TSSC at heights H1 and H2 was significantly higher than that at other heights ($p < 0.001$). Similar results were also reported in *D. asper*, *D. giganteus*, and *D. hamiltonii* [41].

The ash content (AC), considered as a rough estimate of mineral content, decreased as the height increased, although the ANOVA results were not significant. It is partially consistent with studies that have shown a gradual decrease in the mineral content of bamboo shoots as they age [11,40,41,44]. Nutrients such as the AC, Ca, Fe, and Zn (except K) were most responsive at the top part of the bamboo shoots. This was partially consistent with other studies, which found that the mineral content at the tip of bamboo shoots is significantly higher than that at the bottom [45,46]. This is probably due to the growth of apical bamboo shoot components occurring earlier and embodying strong apical dominance, inhibiting nutrient accumulation in the middle and base segments [47]. Interestingly, the concentrations of AC, Ca, Zn, and Fe in the top part from H1 to H3 exceeded those in other parts, but the excess value disappeared at H4 and H5 (Figure 2). We inferred that this is probably due to both the dilution effect of nutrients during rapid shoot growth and the limiting effect of mineral nutrients in the soil [5,48]. Combined with the soil mineral element analysis, it appears that there was a significantly lower Zn and effective K content at the flourishing stage of the bamboo shoots than at the early stage ($p < 0.05$); it may be implied that bamboo shoots have a high demand for the two elements.

### 4.3. The Association between Indicators and Their Effect on Shoot Development

Calcium (Ca) plays a fundamental role in membrane stability, cell wall stabilization, and cell integrity. Potassium (K) is highly mobile in plants and is involved in enzyme activation, photosynthesis, and cell growth [49]. Zinc (Zn) plays a role in the detoxification of superoxide radicals, membrane integrity, and carbohydrate metabolism. Iron (Fe) plays a crucial role in the redox system in cells and in various enzymes [50]. Although there were significant or nearly significant positive correlations among several mineral elements (such as Ca, Fe, Zn, and K) in different parts of the bamboo shoots, it is worth noting that significant correlations between these mineral elements and cellulose and soluble sugar were only detected in the top part of the bamboo shoots (Figure 4). This result suggests that the elements involved in the synthesis and regulation of cellulose and soluble sugar formation may primarily occur in the top part of bamboo shoots. Gene expression in the *D. brandisii* shoots verified that cell cycles and nuclear division are more concentrated in the upper portion of the shoot [38].

Essential mineral elements not only make up the plant's nutrition but also influence its growth and development. It is widely recognized that plants require essential macronutrients such as potassium (K), calcium (Ca), and phosphorus (P), as well as micronutrients like iron (Fe) and zinc (Zn) [51,52]. This study showed that the AC positively correlated with all the mineral elements provided (except for P). The mineral elements Ca, Fe, and Zn are positively correlated with each other, suggesting that these three mineral

elements likely have synergistic and homeostatic roles with each other during shoot development in *D. brandisii*. Numerous studies have shown that metal elements (such as Fe, Zn, Cu, and Mn) interact and mutually influence their presence in plants [53]. A deficiency or excess of any nutrient can cause an imbalance in the uptake of other nutrients, due to their interactions [51]. Thus, it is understandable that there is a consistent change between Fe and Zn levels with the shoot growth of *D. brandisii*.

The PCA results showed that the first principal component (PC1) was mainly contributed to by the AC, Ca, Fe, and Zn. It indicated that the AC and mineral elements were the primary factors affecting the nutrients in the bamboo shoots of *D. brandisii*. However, the impact mainly stemmed from the various parts at the first three height levels. Thus, as a rule of thumb, different parts within the shoots represent the nutrient architecture more than the height grades of the bamboo shoot.

### 5. Conclusions and Limitations

This study assessed the distribution and dynamic changes in essential nutritional components during the growth process of bamboo shoots. The results showed that the CFC and TSSC had a higher content in the lower part of the bamboo shoots, while other nutrients and mineral elements (SPC, AC, P, K, Ca, Fe, and Zn) had a higher content in the upper part. Although no significant differences in the activities of the three metabolic enzymes were detected among different parts and heights of the shoot, the upper part of bamboo shoots showed relatively high activity on average. The content of CFC, TSSC, SPC, P, and K in different parts of the bamboo shoots shows a relatively consistent trend with the growth of the shoot height, while the content of AC, Ca, Fe, and Zn in the upper part of the bamboo shoots shows a similar trend with the growth of the shoot height and shows a completely different trend from that in the middle and lower parts. The above results implied that the shoot tip may be an important tissue for nutrient synthesis during the early growth stage of bamboo shoots (H1-H3). Combining nutrition and taste, the first two periods could be the ideal stages for harvesting bamboo shoots, due to their better taste and richer nutrients.

Although our results give a useful reference for nutrient analysis and a guidance for the bamboo shoot harvesting of *D. brandisii*, it should be pointed out that these results are based on one year of shoot production. Although there is no conclusive evidence in bamboo nutrition studies, many factors could affect mineral absorption and nutrition accumulation (such as a variation in meteorological conditions between years) [54,55]. The analysis of nutrients using bamboo shoots from two or more consecutive years is recommended for future studies in order to obtain more robust results. Future studies into antinutrients (such as cyanogenic glycosides, glucosinolates, tannins, oxalates, and phytates), which are prevalently contained in the shoots of many bamboo species [1,56], should be essential for the shoots of *D. brandisii*.

**Supplementary Materials:** The following supporting information can be downloaded at: https://www.mdpi.com/article/10.3390/horticulturae10050438/s1, Table S1: Content of mineral elements in the root soil of *D. brandisii*.

**Author Contributions:** Conceptualization, W.J. and T.B.; writing—original draft preparation, formal analysis, data curation, L.L.; investigation, Y.W., J.X. and T.B.; supervision, M.Y. and W.J. All co-authors contributed to the discussion, revision, and improvement of the manuscript. All authors have read and agreed to the published version of the manuscript.

**Funding:** This research was funded by the Key Research and Development Program of Guangxi (Grant No. AB21238014) and the National Training Program of Innovation and Entrepreneurship for Undergraduates (No. 202210593026).

**Data Availability Statement:** Data are contained within the article.

**Acknowledgments:** We thank the staff of Guangxi State-Owned Qipo Forest Farm for experimental materials support.

**Conflicts of Interest:** The authors declare no conflicts of interest.

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
