# Peer review of "Different Parts of a Dendrocalamus brandisii (Munro) Kurz Shoot, Rather Than the Shoot’s Height, Are More Indicative of Its Nutrient Properties"

_horticulturae, doi:10.3390/horticulturae10050438_

Round 1
Reviewer 1 Report
Comments and Suggestions for Authors
Dear Authors,
this is a very interesting article about Different parts of a Dendrocalamus brandisii bamboo shoot
The manuscript and the data presented are clear and consistent.
All the sections and additional files are clear and well prepared.
Title, abstract,Introduction and Materials and Methods: I would be grateful if the authors may add the Botanist initial letter at the botanical name of the plants.
Abstract: It could be shortened and explained better
Line 110: Chinese standard ‘GB’: For GB I would be grateful if you would clarify the acronym the first time.
Conclusions: I would be grateful if the authors could add a conclusion that summarizes all the interesting scientific results presented in the article.
References: I would be grateful if the authors may improve the introduction about the chemical composition with this reference:
Curci, Francesca, et al. "Phyllostachys pubescens: from traditional to functional food: Phyllostachys pubescens as functional food." Food Reviews International 39.3 (2023): 1250-1274.
________________________________________________________________________________________________
Although the overall language is good, there are some mistakes, e.g. format/style, grammar/syntax, and/or spelling/typographical errors. The authors would therefore be advised to read through the manuscript carefully in order to eliminate these typos.
Author Response
- Title, abstract, Introduction, and Materials and Methods: I would be grateful if the authors may add the Botanist initial letter at the botanical name of the plants.
A: Thank you for your suggestion. we have corrected the bamboo species’ name as ‘Dendrocalamus brandisii (Munro) Kurz’.
- Abstract: It could be shortened and explained better.
A: Thank you for your suggestion. we have rewritten and improved this part in the revised manuscript (lines 11-13, lines 17-31).
- Line 110: Chinese standard ‘GB’: For GB I would be grateful if you would clarify the acronym the first time.
A: Thank you for your suggestion. The GB is the abbreviation of ‘国家标准Guojia Biaozhun’, which in English is ‘National Standard’. we have revised the ‘GB’ to ‘GB/T, National Standards of the People’s Republic of China’ (lines 325-326).
- Conclusions: I would be grateful if the authors could add a conclusion that summarizes all the interesting scientific results presented in the article.
A: Thank you, we have added the conclusion as the last part. Besides, the limitations in our results and future efforts were also addressed (lines 364-388).
- References: I would be grateful if the authors may improve the introduction about the chemical composition with this reference: Curci, Francesca, et al. "Phyllostachys pubescens: from traditional to functional food: Phyllostachys pubescens as functional food." Food Reviews International 39.3 (2023): 1250-1274.
A: Thank you for your suggestion. we have modified the introduction and the reference was cited, and other several recent valuable references, as well (lines 40, 50,51).
Reviewer 2 Report
Comments and Suggestions for Authors
Dear Horticulturae Editorial Office
I share my observations about the manuscript ID: horticulturae-2874428 - Different parts of a Dendrocalamus brandisii bamboo shoot, rather than the shoots height, are more indicative of its nutrient Properties. The manuscript is a description about macronutrients in bamboo shoots according for height harvest. The study doesn’t have treatments and experimental design. There weren’t scientific findigns.
Best regards!
Author Response
We are very sorry for the reviewer's comments. In spite of this, we have tried our best to revise and improve the manuscript quality, and also explain the shortcomings of the paper in the discussion, and sincerely hope to get the agreement of the reviewer.
Reviewer 3 Report
Comments and Suggestions for Authors
Researchers evaluated a bamboo species for harvesting edible shoots related to the edibility and nutritional value of newly emerging juvenile shoots at five degrees of plant height.
The work addresses an interesting topic but has methodological problems and in-depth discussion that compromises its scientific quality.
The authors would have to improve the introduction to include what has been published on the topic and reinforce the gap and novelty of their research.
The absorption of nutrients by plants and shoots is due to soil fertility and the authors did not indicate chemical analysis of the soil and the fertilizer applied to bamboo. The authors would have to include in the assessment all plant nutrients and also beneficial elements such as silicon, which is considered a silicon-accumulating species.
The authors did not make a discussion that could relate the nutrient levels in the sprout and the maximum levels allowed for humans who will feed on this sprout. There is no point in choosing the part of the sprout that is richest in nutrients but the contents can be toxic to humans. Hence the usefulness of the research is limited.
It is important to have a discussion about the levels of phytoavailable nutrients and not the total content of nutrients present in the sprout and interact with the issue of the presence of antinutrients, which greatly reduces the availability of nutrients for humans.
Author Response
- The work addresses an interesting topic but has methodological problems and in-depth discussion that compromises its scientific quality.
A: Thank you for your comments. We agree that there was a lack of comprehensive thinking in the initial manuscript writing, and based on this, we have made modifications and improvements to the introduction (lines 36-70, 85-89) and discussion sections in the revised manuscript (lines 290-299, 304-309,341-343). On the other hand, this experiment’s materials were randomly sampled in a relatively consistent environment, with three biological replicates for each sample, under the three basic principles of experimental design (randomness, replicates, and environmental control). Therefore, we believe that the results of this study are relatively reliable and have important reference values for bamboo cultivation in forest areas. In addition, we supplemented the soil mineral elements content in the root zone of bamboo shoots during the early and flourishing stages of the experimental site, further enhancing the integrity of this study (lines 126-130, 199-202).
- The authors would have to improve the introduction to include what has been published on the topic and reinforce the gap and novelty of their research.
A: Thank you for your suggestion. We have improved the introduction based on several critical references about bamboo shoot nutrition, and we also further refined the novelty and limits of the past studies (lines 46-61) and emphasized the focus of our work (lines 82-90).
- The absorption of nutrients by plants and shoots is due to soil fertility and the authors did not indicate chemical analysis of the soil and the fertilizer applied to bamboo. The authors would have to include in the assessment all plant nutrients and also beneficial elements such as silicon, which is considered a silicon-accumulating species.
A: Thank you for your constructive suggestion. In fact, we conducted soil sampling in the root zone of bamboo on March 10 and July 30, in the early stage and flourishing stage of bamboo shoots emerging, respectively. According to your suggestion, we have measured and analyzed the several relevant indicators, and discussed the mineral indicators correlation between soil and shoot. In contrast to many other crops in the field, the Dendrocalamus brandisii bamboos in our study were mainly planted under the forest or on the forest edge for agroforestry study and flourished naturally with minimal human intervention, demonstrating an impressive resilience to diseases, pests, and weather‐related challenges, and thrive without the need for fertilizers or pesticides. At the same time, because it is a test forest, except for the initial application of base fertilizer, no fertilizer has been applied in the past 4 years. In the study, ten nutritional indexes determined are some important indexes that are often paid attention to in bamboo shoot research. There is no doubt that silicon is an important mineral element for bamboo growth, and few studies have explored it in bamboo shoots from the perspective of food nutrients. The absorption, accumulation, and distribution of silicon element in D. brandisii will be included in our further study plan.
- The authors did not make a discussion that could relate the nutrient levels in the sprout and the maximum levels allowed for humans who will feed on this sprout. There is no point in choosing the part of the sprout that is richest in nutrients but the contents can be toxic to humans. Hence the usefulness of the research is limited. It is important to have a discussion about the levels of phytoavailable nutrients and not the total content of nutrients present in the sprout and interact with the issue of the presence of antinutrients, which greatly reduces the availability of nutrients for humans.
A: Thank you for your suggestion. We have discussed the safety for consuming the shoots of Dendrocalamus brandisii (lines 309-315). Due to the shoots of D. brandisii empirically recognized as having excellent taste, we have indeed overlooked the determination of some widely mentioned antinutrients in the bamboo shoot. As a limitation, we addressed this concern at the end of the revised manuscript (lines 398-401).
Reviewer 4 Report
Comments and Suggestions for Authors
Review
Manuscript ID: horticulturae-2874428
Title: Different parts of a Dendrocalamus brandisii bamboo shoot, rather than the shoots height, are more indicative of its nutrient properties
Authors: Luxi Li, Jingyun Xu, Yuzhuo Wen, Tiandao Bai, Mei Yang, Weixin Jiang
The research topic is interesting and relevant. The manuscript flows smoothly, with rigorous methodology ensuring reliable and valid results. The presentation of results is articulate and concise, while the discussion provides insightful analysis. The authors conducted this study to evaluate the distribution of nutrient components along the bamboo shoots and how this distribution changes with shoot height growth.
A few notes:
1. A hypothesis is missing in section Introduction. The authors should present a formulated hypothesis, what they hope to reveal with this research and why.
2. It is known that the amount of substances accumulated in the product depends on many factors. One of them is meteorological conditions, which affect the absorption of nutrients from the soil or from fertilizers during the season. Meteorological conditions vary from year to year, and often significantly. Therefore, I have a reasonable question: how reliable is the data of this study if the bamboo shoots were collected and analyzed for only one year. It is likely that next year, when there will be slightly different meteorological conditions, the chemical composition of the bamboos will also be different. Maybe it would be possible to supplement the work with at least one more year's harvest quality analysis.
3. The results are presented concisely and clearly.
4. There is no separate Conclusions section. Authors should formulate clear conclusions.
5. References are not prepared according to the requirements of this journal.
6. 35 references out of 46 are older than five years. I suggest you look at newer literature as well.
Author Response
- A hypothesis is missing in section Introduction. The authors should present a formulated hypothesis, what they hope to reveal with this research and why.
A: Thank you for your valuable comments, and we have improved the introduction and presented the scientific hypothesis (lines 85-86) in the last paragraph of the introduction.
- It is known that the amount of substances accumulated in the product depends on many factors. One of them is meteorological conditions, which affect the absorption of nutrients from the soil or from fertilizers during the season. Meteorological conditions vary from year to year, and often significantly. Therefore, I have a reasonable question: how reliable is the data of this study if the bamboo shoots were collected and analyzed for only one year. It is likely that next year, when there will be slightly different meteorological conditions, the chemical composition of the bamboos will also be different. Maybe it would be possible to supplement the work with at least one more year's harvest quality analysis.
A: First of all, we agree with the reviewer's suggestion. It is true that the accumulation of nutrients in plants is affected by many factors including meteorological conditions. D. brandisii, in the study, is a large tropical and subtropical evergreen sympodial bamboo species with shoots and wood value, and its bamboo shoot quality needed to be evaluated based on field test. Inevitably there are uncontrollable external factors, such as climate, soil and human beings. Based on practical production experience from transplantation experiments, it has been observed that D. brandisii consistently maintains excellent shoot quality wherever it manages to survive, indicating that it is not influenced by environmental factors. Although our results give a useful reference for nutrient analysis and a guidance for bamboo shoot harvesting of D. brandisii, it should be pointed out that these results are based on one year of shoots production. Future studies will carry out nutritional analysis of bamboo shoots from two or more consecutive years, and we expect to obtain interesting findings and robust conclusion. In view of this limitation, we also pointed out at the end of the paper (lines 393-401).
- The results are presented concisely and clearly.
A: According to your suggestion, we have improved the results and analysis.
- There is no separate Conclusions section. Authors should formulate clear conclusions.
A: Thank you for your comment, we have added the Conclusion (lines 379-392).
- References are not prepared according to the requirements of this journal.
A: Thank you, we have corrected this part according to the requirements of this journal.
- 35 references out of 46 are older than five years. I suggest you look at newer literature as well.
A: We have appropriately added the new literature of the last five years to the text.
Reviewer 5 Report
Comments and Suggestions for Authors
Comments in the attached document

Author Response
Remarks
1. Line 96. Soil is “relatively fertile”. Fertile, then, which one? Suitable or unsuitable for growing bamboo? It is worthwhile to give the basic physical and chemical parameters of the soil (table?).
A: Thank you for your comments. We apologize for the imprecise expression, and the statement has been removed from the revised manuscript. In addition, the physical and chemical properties of the soil in this experimental site before bamboo planting have been previously reported (Tang et al., 2022), and we have cited the literature in the revised version (line 98).
Tang M Y, Yang K T, Huang C H, Chen B L, and Jiang W X. Preliminary Study on the Effects of Medicinal Herbs Planting on Soil Nutrients and Enzyme Activities in the Understory of Low Hills[J]. Journal of Central South University of Forestry & Technology, 2022,42(12):142-152, Doi:10.14067/j.cnki.1673-923x.2022.12.015.
2.Line 103. How many shoots were harvested in each group and how many shoots were harvested in total?
A: Thank you for your comments. More than fifty bamboo shoots were harvested in each group by the workers of the Qipo state-owned forest farm. We randomly selected three shoots in each group, totaling 15 shoots with a ground diameter of about 12 cm ~ 14 cm were used as our study materials (lines 110 ~ 112).
3. Line 110. Why were only 3 “representative” shoots for each group collected? According to the authors, is this a sufficient number? Especially since the relationships are studied.
A: Thank you for your comments. The role of the "representative" is undoubtedly crucial in scientific research. However, few studies have explored the relationship between replications and the stability of nutrients in bamboo shoots. We selected three biological replicates (i.e., three bamboo shoots in our study) for each group, a commonly used experimental strategy in many studies (Wei et al. 2019; Zhang et al. 2021; Kofi Bih et al. 2023; GUO Mingyang et al.). Our study demonstrated relatively stable and consistent results across three replications, indicating that using three bamboo shoots in each group is reliable for nutrient and mineral study. In addition, due to the uniform cultivation and management practices, as well as the relatively consistent environmental conditions, we believe that three bamboo plants in each group was sufficient for detecting the nutrient and mineral properties of D. brandisii shoots.
GUO Mingyang, HE Yuelin, PAN Kaiting, et al Analysis of Differential Metabolites of Phyllostachys edulis Shoots at Different Growth Stages by Ultra-high Performance Liquid Chromatography-Tandem Mass Spectrometry. Food Science 44:283–291. https://doi.org/(in Chinese with English abstract) DOI:10.7506/spkx1002-6630-20230207-060.
Kofi Bih F, Antwi K, Appiah-Yeboah J (2023) Mineral Concentration and ash content of bamboo (Bambusa vulgaris Schrader ex Wendland var. vulgaris) culm growth stages in three ecological zones in Ghana. J Bamboo Rattan 21:67–76. https://doi.org/10.55899/09734449.22/21.2b/331
Zhang J, Ma R, Ding X, et al (2021) Association among starch storage, metabolism, related genes and growth of Moso bamboo (Phyllostachys heterocycla) shoots. BMC Plant Biology 21:477. https://doi.org/10.1186/s12870-021-03257-2
Wei Q, Guo L, Jiao C, et al (2019) Characterization of the developmental dynamics of the elongation of a bamboo internode during the fast growth stage. Tree Physiology 39:1201–1214. https://doi.org/10.1093/treephys/tpz063
4. In how many repetitions was the "nutrient and mineral indicators measurement" performed?
A: Thank you for your question. Three biological replications (i.e., three bamboo shoots) were used in each height group, with a total of five height groups in our study.Each bamboo shoot was divided into three parts: top, middle, and base. A total of 9 samples (3 shoots x 3 parts) were collected in each height group for measuring nutrient and mineral indicators.During the measurement process, each sample was repeated three times to calculate the mean value.We have elaborately explained the methods in the Methods section (lines 144-145).
5. Line 126. “ … changes of mineral components in the soil of experiment stand” " What changes were studied? By what methods? In which section of the manuscript were the results of the analyses presented?
A: Thank you for your comments. The essential mineral element content in the soil of the root zoom was compared before and during bamboo shooting using a t-test (supplemental material, Table S1). The soil samples were processed and the mineral contents in the soil were measured according to the standard protocol of the State Forestry Administration of China, and the methods are described in lines 159-172. The results were briefly described (lines 219-222) and discussed (lines 346-349) in the revised manuscript.
6. Lines 314-315. Is this the conclusion of the authors of the study or is it literature data?
A: Thank you for your comments. This is the conclusion of the study. Based on the trends of all nutritional indicators in our study. We concluded that the first two periods (20 ~ 40 cm) could be the ideal stages for harvesting bamboo shoots due to their better taste and richer nutrients.
7. Line 396 „variation of meteorological conditions among years” Is this the conclusion of the authors of the study or is it literature data? The authors did not study the effect of climate on mineral accumulation.
A: Thanks for your comments. Sorry for our imprecise words. Although few studies have explored the effect of meteorological conditions on bamboo shoot growth, drawing from the previous reports on other plants (Elbasiouny et al. 2022; Wang et al. 2023), we speculate that the variation of meteorological conditions over the years may impact mineral absorption and nutrition accumulation. Therefore, we have identified this as a limitation and a potential focus for future studies at the end of the manuscript. We have revised the statement and included relevant literature (lines 399-401).
Elbasiouny H, El-Ramady H, Elbehiry F, et al (2022) Plant Nutrition under Climate Change and Soil Carbon Sequestration. Sustainability 14:914. https://doi.org/10.3390/su14020914
Wang L, Yu X, Gao J, et al (2023) Patterns of Influence of Meteorological Elements on Maize Grain Weight and Nutritional Quality. Agronomy 13:424. https://doi.org/10.3390/agronomy13020424
8. Line 540. What does "additional table" mean? There is already a table with the number 1 in the manuscript.
A: Thank you for your comments. Our non-standard description caused ambiguity, and we have revised the ‘additional table 1’ to Supplemental material ‘Table S1’.
Reviewer 6 Report
Comments and Suggestions for Authors
Comments to the manuscript horticulturae-2874428 "Different parts of a Dendrocalamus brandisii bamboo shoot, rather than the shoots height, are more indicative of its nutrient properties".
Authors propose the report of an experiment aimed to investigate the chemical composition and nutritional properties of the Dendrocalamus brandisii bamboo shoot as influenced by shoot lenght and fragment position (basal, intermediate or apical). The experiment was well designed and correctly carried out. A lot of information was provided on the sample moisture content, crude fiber, total soluble sugars, proteins, ash, phosphorus, potassium, calcium, iron, zinc, as well as activity of neutral invertase, sucrose-phosphate synthase, and ATP synthase. The manuscript is well organized and clearly written. The introduction presents a complete state of the art, the scientific problem, and the research objectives. Material and methods are complete. Results are sound, clearly presented and well discussed. Conclusions are supported by the research findings. The bibliography was cited appropriately. In my opinion the manuscript may represent a good contribute to the field of study and is suitable for publication after some minor changes.
Please check the following suggestions for some minor changes:
1) line 164: please change "... acid-hygrogen ..." with "... acid-hydrogen ..." ;
2) line 241: please change "... corrected ..." with "... correlated ...".
Comments on the Quality of English LanguageMinor editing of English language required.
Author Response
Thank you for the reviewer's comments, and we have revised according to your suggestions.
Round 2
Reviewer 2 Report
Comments and Suggestions for Authors
Dear Horticulturae Editorial Office
I share my observations about the manuscript ID: horticulturae-2874428 - Different parts of a Dendrocalamus brandisii bamboo shoot, rather than the shoots height, are more indicative of its nutrient Properties. The authors made important alterations to the manuscript. However, the study doesn’t have an experimental period, soil type, treatments and experimental design. The manuscript described the results of a sampling of an experimental understory bamboo cultivation. However, a scientific article isn’t only an isolated description because it should be used as the basis for other studies. Thus, it should have a design that is experimental and have details that could possibly be repeated because the study can be used as the basis for other studies. This manuscript has a descriptive role.
Best regards!

Author Response
Thank you for your suggestion. We have corrected some errors according to your comments and reviewed the full text. For the question “What was the experimental period and the environmental conditions (temperature and precipitation) during the period?” (line 97) and “Was this the experimental period?” (line 127), this is mainly due to our lack of clarity. We have added the information for Dendrocalamus brandisii plantation and bamboo shoot harvest time (lines 95-98), and the environmental factors described in this paper are the four years of bamboo growth.
Reviewer 3 Report
Comments and Suggestions for Authors
The authors revised the manuscript and improved the scientific quality and recommended it for publication.
Author Response
Thank you for the reviewer's reply, and thank you for your publication suggestion.